# A Transmission Tower Tilt State Assessment Approach Based on Dense Point Cloud from UAV-Based LiDAR

**Zhumao Lu** [1], **Hao Gong** [2], **Qiuheng Jin** [3], **Qingwu Hu** [3] and **Shaohua Wang** [3,*]

1 State Grid Shanxi Electric Power Research Institute, Taiyuan 030001, China; luzhumao@sx.sgcc.com.cn
2 State Grid Electric Power Research Institute, Nanjing 211106, China; gonghao@sgepri.sgcc.com.cn
3 School of Remote Sensing and Information Engineering, Wuhan University, Wuhan 430079, China; zjqhhh@whu.edu.cn (Q.J.); huqw@whu.edu.cn (Q.H.)
* Correspondence: shwang@whu.edu.cn; Tel.: +86-27-68778562

**Abstract:** Transmission towers are easily affected by various meteorological and geological disasters. In this paper, a transmission tower tilt state assessment approach—based on high precision and dense point cloud from UAV LiDAR—was proposed. First, the transmission tower point cloud was rapidly located and extracted from the 3D point cloud obtained by UAV-LiDAR line patrol. A robust histogram local extremum extraction method with additional constraints was proposed to achieve adaptive segmentation of the tower head and tower body point cloud. Second, an accurate and efficient extraction and simplification strategy of the contour of the feature plane point cloud was proposed. The central axis of the tower was constrained by the contour of the feature plane through the four-prism structure to calculate the tilt angle of the tower and evaluate the state of the tower. Finally, the point cloud of tower head from UAV-based LiDAR was accurately matched with the designed tower head model from database, and a tower head state evaluation model based on matching offset parameters was proposed to evaluate tower head tilt state. The experimental results of simulation and measured data showed that the calculation accuracy of the tilt parameters of transmission tower body was better than 0.5 degrees, that the proposed method can effectively evaluate the risk of tower head with complex structure, and improve the rapid and mass intelligent perception level of the risk state of the transmission line tower, which has a wide prospects for application.

**Keywords:** UAV LiDAR; point cloud; transmission tower; tilt; state assessment

## 1. Introduction

The demand for energy is increasing, with the rapid development of economies. As a kind of basic energy, electric power resources play an extremely important role. In order to alleviate the problem of insufficient electricity power, China's national electric power system is also developing at a higher speed. In terms of infrastructure construction, such as wire networks, long-distance power supply construction of high-voltage and ultra-high-voltage transmission lines in depopulated areas is the top priority, and higher requirements are put forward for the safe operation of the power system in depopulated areas [1–3]. Transmission line routing consists of transmission lines, towers, and their accessories. Towers are the basic equipment for transmission lines. Weather factors or terrain conditions may cause transmission line wear, corrosion, broken strands, and other damages, resulting in deflection or collapse of towers [4,5]. In addition, when transmission towers are installed and fixed, the selection of construction location can cause the tilt deformation of the tower. Various construction and excavation activities around the power line can also aggravate the occurrence of geological disasters, easily causing damage to the tower itself and the ground in the construction area, and cause the tower tilt and short circuit fault in the transmission line, all of which seriously threaten the safe operation of the power line [6,7]. Tower tilt can easily cause power line pulling, sagging, and other problems. If the above serious

consequences are not repaired and checked in time, major accidents will be caused, which will lead to regional power outages and inestimable losses [8,9]. Therefore, it is necessary for transmission line management units to grasp these changes in time and evaluate and predict line safety under various conditions.

Traditional methods for measuring the inclination of line towers mainly include the plumb method, theodolite method, and the plane mirror method. These methods are not only difficult to implement but also inefficient for the safe operation and development of the modern power grids [10–12]. Online monitoring sensors and optical fiber sensors are usually the professional instruments used for tilting measurement, which can be installed on the tower to determine whether serious tilting occurs and send the monitored tilting warning information to the operation and maintenance management department in time through GPRS (General Packet Radio Service) or mobile signal [13–16]. However, the installation, power supply, and data transmission of sensors are restricted by various factors, so it is difficult to promote their large-scale application.

With the development of the smart power grid in China, intelligent digital processing technology has become widely used in the power industry [17–19]. As a new generation of remote sensing technology, UAV LiDAR technology takes laser pulse as a measurement medium and highly integrates advanced equipment such as GPS, INS, and laser scanning rangefinder, which can quickly obtain high-precision 3D coordinates of targets and high-precision 3D laser point cloud data of transmission corridors [10,20]. By accurately measuring the tilt rate of the transmission tower point cloud, the defamation law can be analyzed and studied, providing accurate quantitative analysis, and providing a flexible way to measure the tilt state of the tower in the transmission line corridor [21–25]. The detection of tower tilt by UAV-carrying camera or laser radar mainly involves the identification of the tower and the detection of tower tilt [26–28]. In terms of image-based tower recognition, Zhang P. et al. [29] and Yan L. et al. [30] analyzed the contour information of the tower using the constant false alarm rate technique combined with the extended fractal technique to identify the target of the towers based on SAR image data. However, this algorithm can only locate the tower area in the radar image; it cannot judge its attitude. Sampedro et al. [31] obtained foreground-background classifiers and type classifiers through supervised training methods. This method can perform well in recognition and classification, but its scope of application is limited. The introduction of deep learning for tower recognition and classification will determine the direction of future research [32].

The laser scanning sensor carried by UAV LiDAR has high density and high penetration ability, which can reduce the influence of the surrounding environment on the shielding of tower targets and realize the omnidirectional data acquisition of transmission towers. The laser point cloud data obtained by UAV LiDAR contains rich spatial location information, which can realize the direct positioning of tower targets. It provides convenience for the calculation of the parameters and the inclination evaluation of the tower target itself. Li et al. [33] found that the filtered tower point cloud was detected through the density and elevation changes of the point cloud after vertical projection, and the position of the tower point cloud was identified in their study. Han W. et al. [34] and Zhang et al. [35] first extracted power line point clouds, and then located and extracted towers by detecting the connection points of power line pairs. Guo et al. [36] first used the seed point growth of point cloud data to extract the tower target and then calculated the precise value by the least square method. Yin H. et al. [37] classified point clouds by using the elevation histogram of point clouds in local blocks to distinguish ground, line, tower, and other elements. Peng X. et al. [38] located the orientation of transmission towers according to the point cloud density in the transmission corridor, the height difference of ground objects at the same horizontal position, and the slope characteristics of ground objects edges.

The tower tilt measurement is based on transmission tower recognition from images and point cloud. Tan et al. [39] used the ground 3D laser scanner to monitor the deformation of the cooling tower in a power plant. By comparing the spatial model collected by the

scanner with the initial design model, the thickness and deformation of the cooling tower in the power plant are obtained. Shen X. et al. [40] realized the measurement of tower tilt of transmission lines through a new method based on the 3D terrestrial laser scanner and carried out field feasibility and comparative tests based on the established test system. Considering the real deformation of towers, Zhao X. et al. [41] proposed an improved point cloud registration algorithm for towers with constraints based on the traditional Iterative Closest Point (ICP) algorithm and successfully applied it to tilt detection of the tower. Wang Y. et al. [42] first extracted the main outline of the tower. Then, the central axis of the tower was extracted according to the contour and the reference direction of the ground normal was selected, and the inclination of the tower was judged by their included angle.

However, the tilt state of the transmission tower is different in different parts. The transmission tower structure is usually composed of tower head (Figure 1a), tower body (Figure 1b), and tower foot (Figure 1c), according to design parameters. Generally, a transmission tower has four inverted triangular pyramid tower feet. The size and elevation distribution of tower feet may be different due to the influence of topography, and the tower feet have little influence on the tilt deformation of the tower. The tower body is shaped like a regular four-prism platform (Figure 1b) with a hollow structure inside. The regular tower body is the main component for calculating the tilt parameters of the tower, and the tilt parameters are usually calculated by fitting the central axis of the tower. However, the relationship between the vector shape and the topological structure of the tower head is complex. In general, tower heads vary considerably from model to model, while the number of models is limited, and the size and structure of the same model are consistent. Based on the above, the tilt parameters need to be calculated separately for the different parts of the tower.

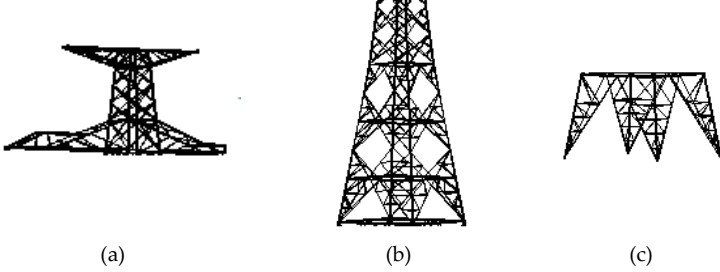

(a)    (b)    (c)

**Figure 1.** Structure map of transmission tower. (**a**) tower head; (**b**) tower body; (**c**) tower foot.

This paper proposes a new tilt state evaluation strategy, based on the segmentation of tower head and body for given differences of tower geometry. First, the transmission tower point cloud is rapidly located and extracted from the 3D laser point cloud obtained by line UAV patrol. A robust histogram local extremum extraction method with additional constraints is proposed to realize adaptive segmentation of the point cloud at the tower head and tower body. Second, an accurate and efficient extraction and simplification strategy of the outer contour of the feature plane point cloud is implemented. The central axis of the tower was constrained by the outer contour of the feature plane through the four-prism structure to calculate the tilt angle of the tower body and to evaluate its state. Finally, the point cloud of tower head from UAV-based LiDAR was accurately matched with the designed tower head model from a database, and a tower head state evaluation model based on matching offset parameters was proposed to evaluate tower head tilt state.

## 2. Methodology

The proposed method in this paper is based on the accurate analysis of the geometric structure of transmission towers. Different calculation methods are adopted for different parts of the tower head and tower body to calculate parameters, so as to achieve a more comprehensive and accurate state assessment of transmission towers. The overall method flow is shown in Figure 2.

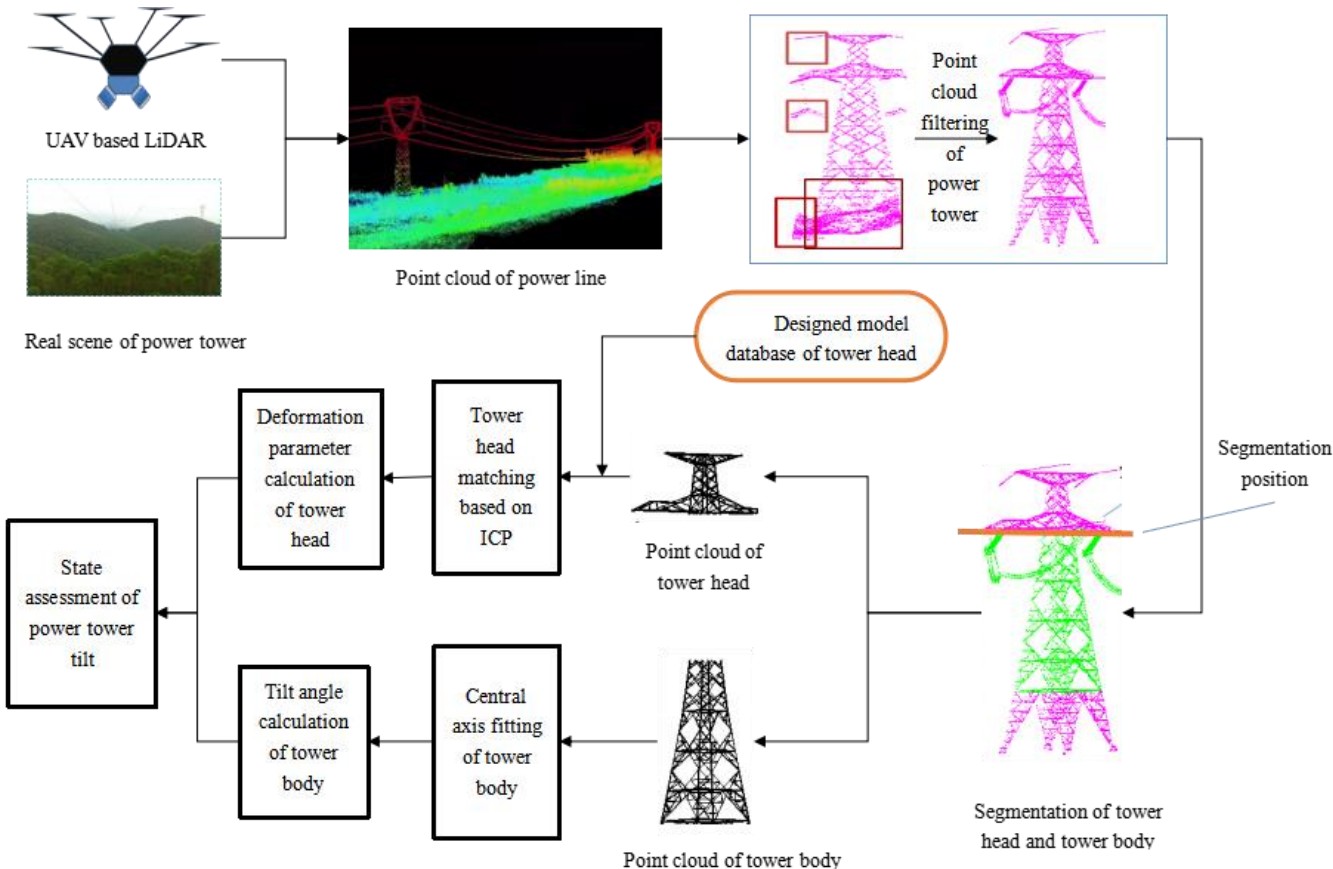

**Figure 2.** Flow chart of the proposed method.

As shown in Figure 2, first, the transmission tower point cloud was extracted by rapid positioning from the point cloud obtained by power line patrol with UAV-LiDAR. Based on the local structure analysis of the point cloud elevation histogram, the tower head and tower body point cloud were adaptively separated. Second, the tower central axis of the four-prism structure was fitted based on the point cloud of the tower body, and then the tilt angle of the tower was calculated, so as to evaluate the tilt state of the tower. Finally, the tower head point cloud was classified according to the designed tower head model database, and the tower head tilt state was evaluated by accurately matching offset parameters between the tower head point cloud and the standard model.

### 2.1. Tower Point Cloud Extraction and Filtering

Point clouds of transmission towers were first positioned and extracted from the whole UAV-LiDAR point cloud data set. Next, point cloud filtering was carried out for a single tower to remove the noise points around the tower.

#### 2.1.1. Tower Positioning and Extraction

As there were some noise points in the original point cloud obtained by UAV LiDAR, the first step was to extract the transmission tower data from the original data using filtering algorithms. In this paper, Kalman filter, one of the most common filters, was adopted to remove the noise points, and to obtain the point cloud of the transmission power corridor, including towers, wires, ground and surrounding vegetation. Among these, transmission tower point clouds were continuously distributed in elevation, that is, point clouds were distributed from the highest point to the lowest point, while cable, vegetation and other point clouds were only distributed within a certain elevation range. Based on these distribution characteristics, this paper proposed a fast positioning and segmentation algorithm based on elevation projection local maximum detection to segment

a single tower point cloud from the complete transmission power corridor point cloud. The specific steps are as follows:

(1) The 3D laser point cloud of the power corridor was projected to the horizontal XOY coordinate system. The projection plane was further divided into a grid of a specific size, and the grid position of each point cloud was determined based on Equation (1):

$$\begin{aligned} m &= (y - y_{min})/d \\ n &= (x - x_{min})/d \end{aligned} \tag{1}$$

where $(x, y)$ are the $XY$ coordinates of point cloud. $x_{min}$, $y_{min}$ are the minimum values of $X$ and $Y$ coordinates of the point cloud. $d$ is the size of grid. $m$ and $n$ are corresponding grid numbers.

(2) Local elevation maximum, minimum, and elevation difference of the projection grid were calculated. Transmission towers were characterized by large elevation differences, so the elevation difference can be used to eliminate the grid area that contained non-transmission tower points such as ground and low vegetation. The elevation difference threshold $\theta_h$ was set to determine grids with greater elevation difference than $\theta_h$.

(3) The point cloud was extracted with the point with the local maximum elevation as the center and the grid size as the radius. The minimum elevation within the extraction area was considered to be the ground elevation and was set as the threshold $H_{min}$ to remove ground points and extract the tower point clouds.

### 2.1.2. Tower Point Cloud Filtering

The point clouds extracted in the above ways inevitably contained non-transmission tower point clouds, such as noise points of the power transmission line on top of the tower, weed points on the bottom of the tower, ground points, etc. Ground points and vegetation points generally gathered near the bottom of the tower. The noise points of power transmission lines near the tower body were generally at a certain distance from the tower body and far from the contour edge of the tower body, while the tower head was directly connected to the power transmission line and the power lines were twined and shuttled, which was complicated and difficult to completely remove (as Figure 3 shows).

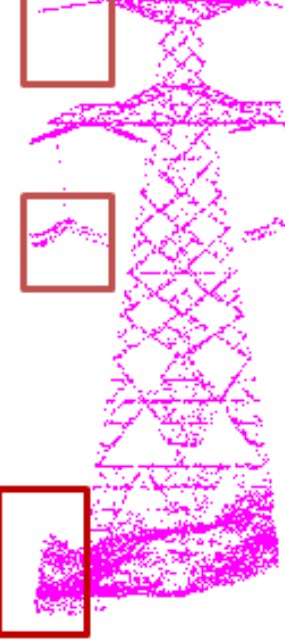

**Figure 3.** Noise points of transmission tower.

As shown in Figure 3, the noise points in the point cloud of a single tower varied greatly in different parts of the tower. Therefore, this paper adopted different filtering strategies for noise points in different parts: (1) for the noise points near the bottom of the tower, we set an elevation threshold $T_h$ ($T_h = H_r + C$, where $H_r$ is the elevation minimum value of the point in the transmission tower point cloud, and $C$ is a constant and is set according to the structure of the transmission tower). The point clouds with elevations less than $T_h$ were rejected to roughly eliminate the noise points at the bottom of the tower. (2) For the transmission tower line points near the tower head and body, the point cloud was filtered from the bottom up by using extracted information through a progressive iterative filtering strategy. Specifically, by calculating the distance between the tower center and point cloud, some of the noise points can be eliminated. Then a part of transmission tower information was extracted and the point cloud was filtered until the desired transmission tower was extracted.

### 2.2. Tower Feature Extraction

A steel beam appeared at every other elevation, as shown in purple in Figure 4. The steel beam that separated tower head and tower body was called the tower shoulder. In general, these steel beams would gather more point clouds and the spacing between each steel beam was greater than a certain threshold (these were referred below as feature planes, and the corresponding elevations as feature elevations).

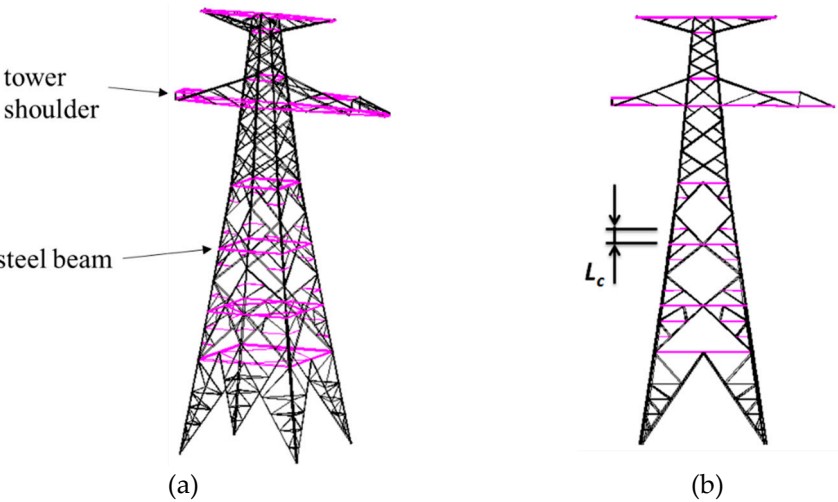

(a)    (b)

**Figure 4.** Feature planes of tower body. (**a**) tower structure; (**b**) interval between feature plane.

### 2.2.1. Feature Elevations and Planes Extraction

Based on the elevation histogram of point clouds, the feature elevations and corresponding feature planes can be extracted. The specific algorithm steps were as follows:

(1) The elevation interval $\Delta H$ was set and a horizontal projection was made based on it, i.e., every point within the elevation range of $\Delta H$ was counted as the point of the same elevation and the histogram of elevation distribution of the point cloud was generated (Figure 5a). Elevation interval $\Delta H$ was the key parameter of structural feature extraction. If the value was too small, the algorithm was inefficient; if the value was too large, the error increased. Considering the width of the transmission tower transverse structure, this parameter was set as 0.1 m in this paper.

(2) $N_{top}$ local maximum $N_{top}$ values were calculated and their corresponding elevation values were regarded as candidate feature elevations. To ensure that no characteristic elevation is omitted in the selection process, the value of $N_{top}$ should be greater than the actual number of steel beams, which is usually smaller than 20. To reduce the possibility of omission, the value of $N_{top}$ was set as 50 in this paper.

(3)    According to the constraint of the minimum interval of the feature plane, the moving window method was adopted to remove the maximum values that were too close to each other. Specifically, a window of size $L_c$ was used to mark the maximum values in the window from low to high (forward) and from high to low (backward) respectively. The same values in the two results were selected as the feature elevation, as shown in Figure 5b.

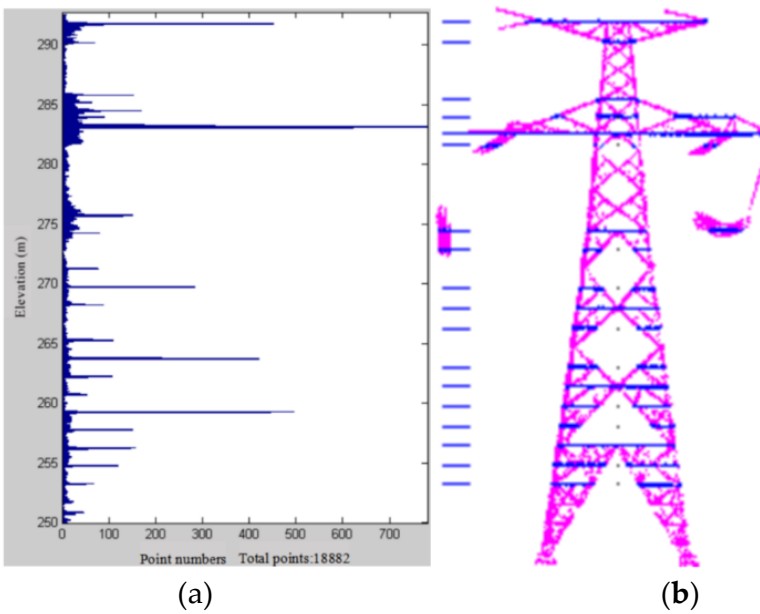

**Figure 5.** Feature plane extraction based on the elevation histogram. (**a**) histogram of elevation distribution; (**b**) feature plane extraction result.

### 2.2.2. Tower Head and Tower Body Point Cloud Segmentation

In order to ensure the stability of the tower, according to the principle of construction stability, there are more steel frame transverse structures at the junction of tower head and tower body, so there were also more point clouds in the corresponding elevation, as shown in the red line in Figure 6.

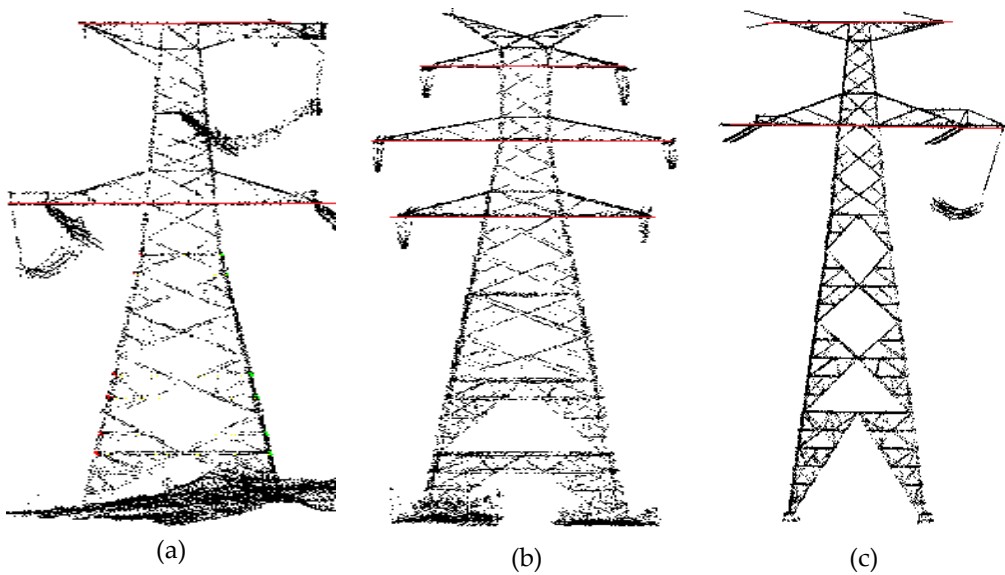

**Figure 6.** Point cloud and structure of different types of tower. (**a–c**) represent three typical towers.

As shown in the Figure 6, there are some long steel beams in the tower head section, which could be easily confused with the elevation of the tower shoulder. Based on the extraction of feature elevations, we proposed a method to extract the elevation of tower shoulder:

(1) The mean of number of points in all feature elevation planes was calculated.
(2) The number of points corresponding to each feature elevation was calculated from the $N_{top}/2$ feature elevation upwards. The first feature elevation where the number of points was greater than the mean was the elevation of tower shoulder, based on which the tower head and tower body was segmented, as shown in Figure 7.

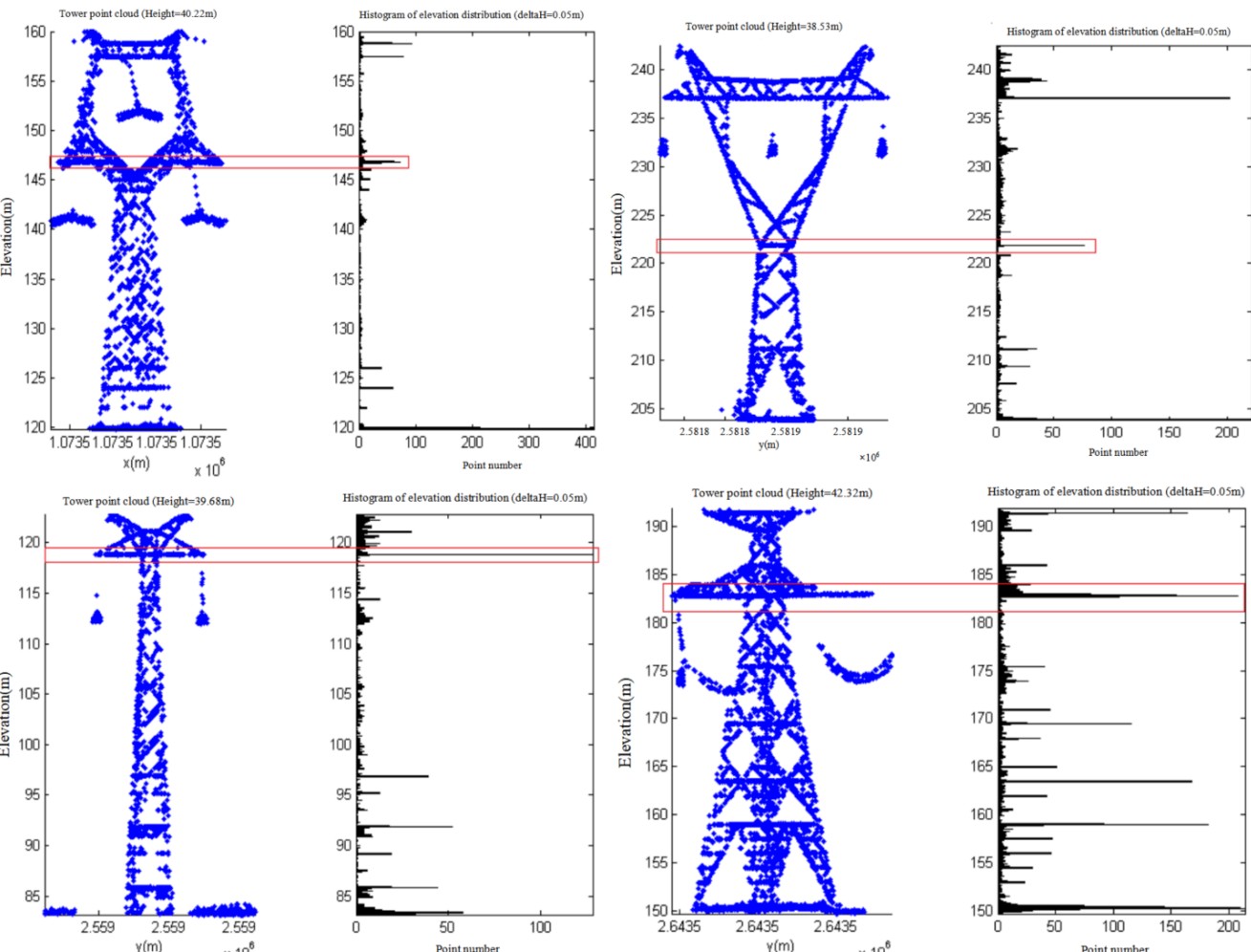

**Figure 7.** Segmentation based elevation of tower head and tower body.

### 2.2.3. Feature Plane Contours Analysis

The Alpha Shapes algorithm was first proposed by Edelsbrunner and is widely used to extract contour lines from point sets [43–45]. For each segmented feature plane point cloud within the tower body elevation range, the Alpha Shapes algorithm was first used to extract the contour of the point cloud. The original contour edge segments of the feature plane extracted by Alpha Shapes were relatively broken, with a wide range of azimuth variations and jagged edges on the same line, which were not easy to orient. Therefore, in this paper, the Pipeline algorithm [46] was then adopted to simplify the original contour and to extract the skeleton line of the contour.

It should be pointed out that power line point clouds generally existed in the multi-layer feature plane point clouds of the tower body obtained by rough segmentation. When the Alpha Shapes algorithm was used to extracted the outer contours, these power line

point clouds had a great impact and had to be removed. In this paper, the contour was extracted from the lowest feature plane, and the extracted minimum contour was then used to limit the range of the next feature plane to be processed, so as to eliminate the power line clutter within the range of the tower body and obtain the final Alpha Shapes contours (as Figure 8b shows).

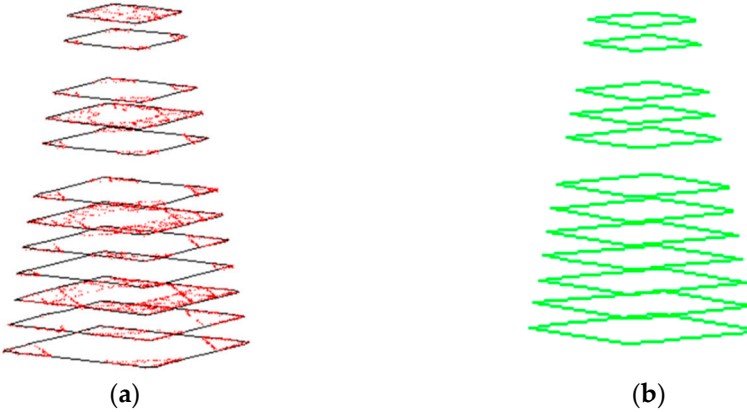

           (**a**)                                           (**b**)

**Figure 8.** Feature plane contours analysis, (**a**) original feature planes; (**b**) simplified feature planes.

### 2.3. Tower Body Tilt Evaluation

The tower tilt was calculated according to the central axis of the tower. Since the tower is a pyramid structure, in order to fit the central axis accurately and robustly, the multi-layered prismatic characteristic plane contour should be extracted from the point cloud of the tower body.

#### 2.3.1. Central Axis Fitting Based on Multi-Layer Feature Planes

The central axis of the transmission tower was obtained by extracting the center of the feature plane of each layer with the simplified contour line. However, the feature extraction and contour analysis of tower structure in Sections 2.2.1 and 2.2.3 were both based on the premise that the tower was not tilted. When the tower was tilted, the feature plane could not be accurately extracted according to the point cloud elevation histogram, thus affecting the subsequent calculation results. In this paper, iterative fitting method was used to obtain the accurate angle between the initial central axis of the tower and X and Y axes by repeatedly calculating the central axis. The specific algorithm steps were as follows:

(1) The angle threshold $\theta_q$ was set. When the calculated angle was less than $\theta_q$, the tower axis was considered vertical. The maximum iteration number iteration$_{\max}$ was set.

(2) The minimum circumcircle of the each feature plane contour was calculated [14], and the center of the circumcircle was taken as the center of the contour. The centers of all circumcircle were connected as the initial central axis (Figure 9).

(3) The central axis was projected onto the YOZ plane and the angle $q_1$ between the central axis and the *X*-axis was calculated (Figure 10a).

(4) When $q_1$ was greater than $\theta_q$, it indicated that the tower rotated around the *X*-axis. Otherwise, skip Step (5).

(5) The point cloud of transmission tower was rotated $q_1$ in the reverse direction around the *X*-axis, and the feature plane extraction and contour analysis were repeated to obtain the new central axis of the tower. Steps 2–4 were repeated until the angle was less than $\theta_q$ or the number of iterations reached iteration$_{\max}$.

(6) The central axis was projected to XOZ plane (Figure 10b), and the angle $q_2$ between the central axis and *Y* axis was calculated.

(7) When $q_2$ was greater than $\theta_q$, it indicated that the tower was rotated around the *Y*-axis. Otherwise, Step (8) was skipped.

(8) The point cloud of transmission tower was rotated $q_2$ in the reverse direction around the *Y*-axis, and the new central axis of the tower was obtained by repeated feature

plane extraction and contour analysis. The central axis was projected to XOZ plane, and the angle between the central axis and the *Y*-axis was calculated again until the angle was less than $\theta_q$ or the number of iterations reached iteration$_{max}$.

(9) The sum of $q_1$ in Step (5) was $q_x$, the angle between the initial central axis and *X*-axis; the sum of $q_2$ in Step (8) was $q_y$, the angle between the initial central axis and the *Y*-axis.

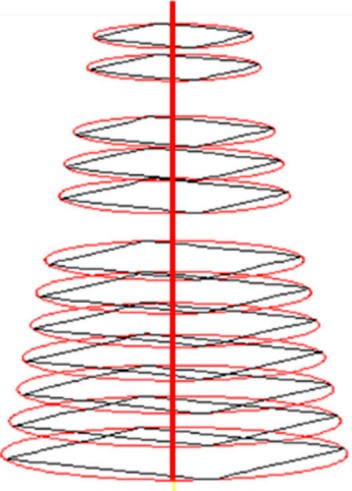

**Figure 9.** Central axis.

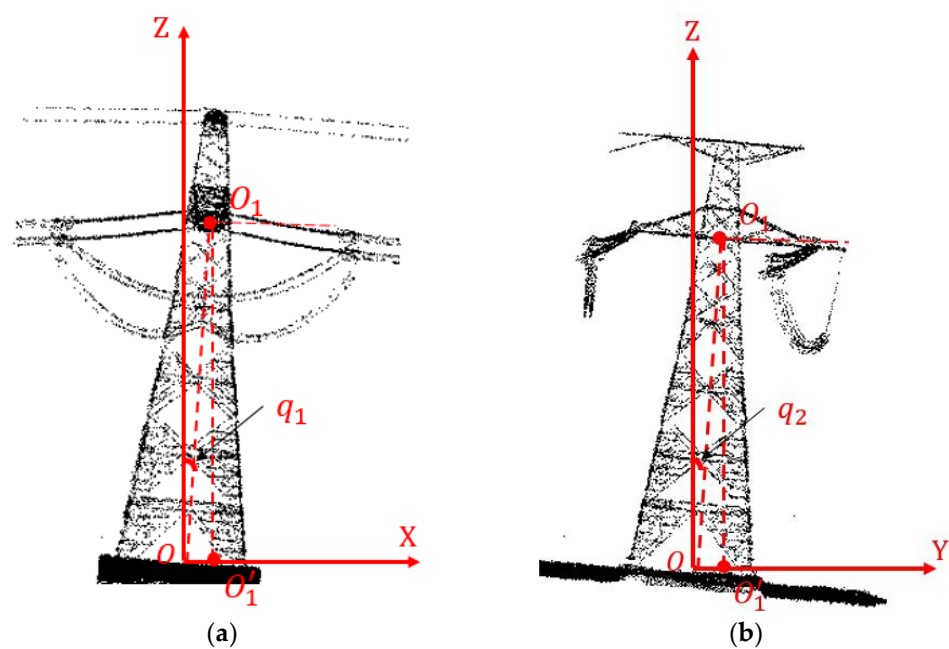

(a)            (b)

**Figure 10.** Central axis projection. (**a**) projection on YOZ plane; (**b**) projection on XOZ plane.

### 2.3.2. Tilt Angle Calculation Based on the Central Axis of Tower

The tower body tilt was determined by calculating the angle between the fitted central axis and the theoretical central axis of the tower (the vertical line of the ground on which the tower was located), i.e., the residual angle between the three-dimensional space vector and the plane Z = 0. Using the coordinates of two points on the vector $O(x_1, y_1, z_1), O_1(x_2, y_2, z_2)$,

the angle between the vector and the plane Z = 0 can be calculated by Equation (2), as shown in Figure 11.

$$Angle = \left| atan\left( \frac{\sqrt{(x_2 - x_1)^2 + (y_2 - y_1)^2}}{|z_2 - z_1|} \right) \right| * 180/\pi \tag{2}$$

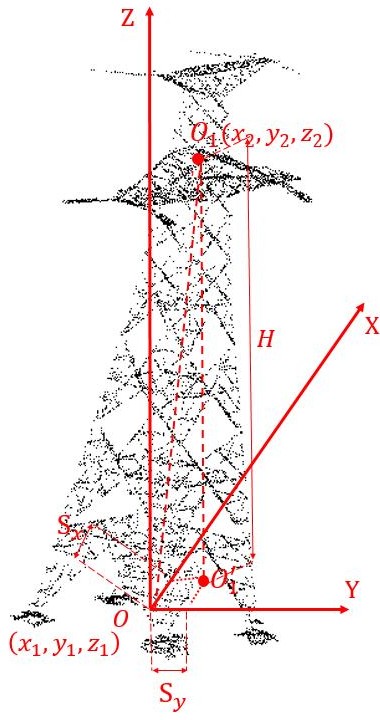

**Figure 11.** Schematic diagram of tower tilt angle calculation.

The *Angle* was used as the basis to preliminarily judge the tower tilt condition. When a threshold $\theta_1$ was set, if the *Angle* was less than the threshold, it can be considered that there is no tilt within the allowable range. If the angle was greater than the threshold, the tower was judged to be tilted, and the tilted angle was the *Angle*.

If the tower is tilted, the degree of tilt can be further judged by another threshold $\theta_2(\theta_2 > \theta_1)$ is set. If $\theta_2 > Angle \geq \theta_1$, the tower is suspected to be tilted, and a patrol should be dispatched to investigate and verify the situation on site. If $Angle \geq \theta_2$, the tower is seriously tilted, and maintenance personnel should be dispatched immediately to repair it.

### 2.4. Tower Head Tilt Status Evaluation

As the structure of the tower head was complex, we adopted the method of matching the tower head to be measured and the corresponding standard tower head model to calculate the distance between the corresponding points, so as to obtain the deformation degree of each part of the tower head. Assuming that the standard tower head model was *A* (i.e., the standard tower head point cloud in the tower head model library) and the tower head point cloud to be detected was *B*, the ICP matching algorithm was adopted to obtain the aligned tower head to be tested, *B*ı, by calculating the optimal rotation matrix *R* and translation matrix *T*. Theoretically, if the transmission tower head was neither tilted or deformed, the point cloud of the tower head to be detected would be fully matched with the standard tower head model, but for a tower head with deformation, there would be a shift. In this paper, Euclidean distance and Hausdorff distance were used to describe the offset deformation, so as to evaluate the deformation state of the tower head. The specific steps were as follows:



(1)    Calculation of tower head deformation offset

For the deformation offset of points in $B'$, by calculating the Euclidean distance between that point and each point in $A$ one by one, the nearest point was the offset point corresponding to that point and the distance between them was the corresponding offset $D_E$, as shown in Formulas (3)–(5):

$$D_E(a, B') = \min_{b_i \in B'} \left( \sqrt[2]{(x_a - x_{b_i})^2 + (y_a - y_{b_i})^2 + (z_a - z_{b_i})^2} \right) \tag{3}$$

where $a(x_a, y_a, z_a)$ was the 3D coordinate of a point in $A$, and $b_i(x_{b_i}, y_{b_i}, z_{b_i})$ was the 3D coordinate of a point in $B'$.

In order to facilitate the deformation analysis of the tower head, the Hausdorff distance was introduced. The minimum Euclidean distance $D_E$, from each point on standard tower head $A$ to tower head $B'$ after registration, and the one-way Hausdorff distance $h$, from standard tower head $A$ to tower head $B'$, were calculated. The minimum Euclidean distance at each point was normalized, as shown in Equation (4). The ratio $D_H$ was the evaluation standard of $B'$ offset of tower head under test. The closer $D_H$ is to 1, the greater the deformation degree of tower head; the closer $D_H$ is to 0, the smaller the deformation degree of tower head.

$$D_H(a, B') = \frac{D_E(a, B')}{h(A, B')} \tag{4}$$

(2)    Overall deformation evaluation of tower head

For the overall deformation of tower head, the ratio of $D_{Avg}$, the average value of $D_H$ of all point clouds to $h(B, A)$, was used and expressed as follows:

$$Tower\ head\ overall\ risk\ status = \begin{cases} Low\ risk & 0 \leq \frac{D_{Avg}}{h(B,A)} < 0.5 \\ High\ risk & 0.5 \leq \frac{D_{Avg}}{h(B,A)} < 1 \end{cases} \tag{5}$$

At the same time, the ratio of the safety area, low-risk area, and high-risk area in the mid-point cloud of tower head was calculated to determine which area in the tower head point cloud had the highest ratio. Combining the two, the overall deformation of tower head was evaluated. The evaluation criteria are shown in Table 1.

**Table 1.** Evaluation standard for integral deformation of tower head.

| $\frac{D_{Avg}}{h(B,A)}$ | The Region with the Highest Proportion | Global Deformation Assessment |
|---|---|---|
| 0–0.5 | The safety area | Low risk and safe |
| 0–0.5 | Low risk area | Low risk and to be detected |
| 0–0.5 | High risk area | Low risk and need required |
| 0.5–1 | The safety area | High risk and safe |
| 0.5–1 | Low risk area | High risk and to be detected |
| 0.5–1 | High risk area | High risk and need required |

## 3. Experiments and Results

### 3.1. Dataset

In the paper, the feasibility of the proposed method was verified using the 3D laser point cloud data provided by Shanxi Power Grid Company. The data were collected using the UAV equipped with a VUX-1 LiDAR sensor. The experimental data set contained 157 transmission towers of six categories, as shown in Table 2. The point cloud density was about 100 pts/m$^2$. The number of point clouds varied considerably across the six types of transmission towers, with the minimum point being 4000 and the maximum point being 40,000 and the average point being 20,000.

**Table 2.** The number of six types of transmission towers.

| ID | Number | Point Cloud Front View (Size Inconsistency) |
|----|--------|---------------------------------------------|
| 1 | 32 | |
| 2 | 26 | |
| 3 | 25 | |
| 4 | 36 | |
| 5 | 19 | |
| 6 | 19 | |

In the data set, most of the transmission towers were vertical, and real tilted towers were rare. To verify the accuracy of the method for calculating the tower body tilt, simulations were carried out on the collected sample data to obtain towers with different tilt angles. Specifically, 10 towers were randomly selected from each category, and the simulated data were divided into six groups, each of which took the base center as the origin and rotated a certain angle around a different axis, as shown in Table 3.

**Table 3.** Tower tilt experiment design.

| Angle | *X* Axis | *Y* Axis | *X* First and then *Y* | *X* Axis | *Y* Axis | *X* First and then *Y* |
|-------|----------|----------|------------------------|----------|----------|------------------------|
| Rotation Angle | 5° | 5° | 5° | 10° | 10° | 10° |
| Tilt Angle | 5° | 5° | 7.066° | 10° | 10° | 14.106° |

The tower head had slight deformation due to uneven force caused by power line pulling and influence of wind. Therefore, the tower head in the raw data was not processed and was used directly as the experimental data set for the tower head tilt and deformation analysis.

### 3.2. Analysis of Tower Body Tilt State Evaluation

The central axis of the tower body was fitted according to the method described in this paper, and the inclination of the tower was judged by calculating the included angle between the central axis and the theoretical central axis. As shown in Figure 12, the red and black lines were the actual central axis and the theoretical central axis of the tower.

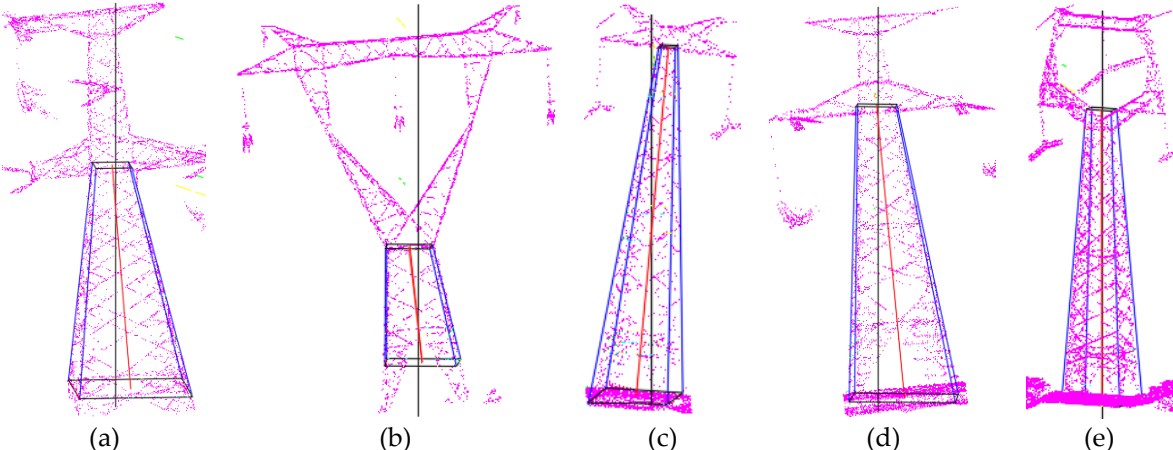

(a)  (b)  (c)  (d)  (e)

**Figure 12.** Central axis calculation result of typical towers. (**a**–**e**) represents five typical types structure of towers.

The tilt of 60 transmission towers was calculated, and the rotation angles of each transmission tower without manual tilt, 5 degrees rotation around $X$ axis, 5 degrees rotation around $Y$ axis, 5 degrees rotation around $Y$ axis, 10 degrees rotation around $X$ axis, 10 degrees rotation around $Y$ axis, 10 degrees rotation around $Y$ axis and 10 degrees rotation around $Y$ axis and the corresponding precision were calculated. The calculation results of 18 transmission towers of six types were randomly selected for analysis, as shown in Table 4. The calculation of transmission tower tilt angle was accurate, and the relative error was better than $0.7°$.

**Table 4.** Calculation accuracy of tower tilt [1].

| Tower Type | No Tilt | X-5° | Y-5° | XY-5° | X-10° | Y-10° | XY-10° | Relative Error |
|---|---|---|---|---|---|---|---|---|
| 1 | 0.159 | 5.577 | 5.209 | 6.838 | 11.119 | 10.938 | 12.964 | 0.630 |
| 1 | 0.135 | 5.77 | 3.821 | 6.684 | 10.441 | 9.634 | 15.83 | 0.710 |
| 1 | 0.258 | 5.27 | 5.116 | 7.393 | 9.929 | 10.349 | 15.995 | 0.463 |
| 2 | 0.282 | 5.882 | 4.91 | 7.324 | 11.036 | 11.056 | 13.748 | 0.570 |
| 2 | 0.321 | 4.854 | 3.78 | 7.569 | 10.109 | 10.377 | 14.446 | 0.425 |
| 2 | 1.193 | 5.282 | 5.253 | 6.769 | 10.888 | 8.507 | 12.931 | 0.803 |
| 3 | 0.04 | 4.298 | 4.847 | 7.241 | 12.543 | 10.919 | 13.627 | 0.720 |
| 3 | 0.915 | 7.051 | 4.922 | 9.981 | 10.535 | 11.356 | 13.586 | 1.200 |
| 3 | 0.229 | 5.202 | 4.77 | 7.399 | 11.079 | 9.116 | 15.265 | 0.582 |
| 4 | 0.235 | 5.498 | 5.228 | 6.348 | 10.054 | 10.453 | 12.706 | 0.518 |
| 4 | 0.181 | 4.793 | 4.852 | 8.125 | 10.55 | 9.326 | 12.784 | 0.596 |
| 4 | 0.219 | 4.695 | 4.309 | 7.277 | 10.16 | 10.372 | 14.444 | 0.322 |
| 5 | 0.065 | 5.172 | 4.11 | 7.515 | 10.879 | 10.457 | 12.959 | 0.584 |
| 5 | 0.183 | 4.963 | 4.496 | 7.918 | 8.675 | 8.798 | 13.653 | 0.655 |
| 5 | 0.272 | 4.832 | 5.591 | 7.432 | 10.587 | 9.856 | 11.814 | 0.636 |
| 6 | 0.136 | 5.535 | 5.007 | 7.975 | 10.712 | 12.154 | 14.786 | 0.728 |
| 6 | 0.283 | 6.342 | 4.234 | 8.229 | 11.111 | 10.367 | 15.406 | 0.899 |
| 6 | 0.081 | 4.629 | 4.539 | 6.842 | 11.307 | 10.382 | 12.319 | 0.665 |
| Relative error | 0.288 | 0.529 | 0.434 | 0.629 | 0.806 | 0.775 | 1.092 | 0.650 |

[1] These values are in degrees.

As shown in Figure 13, in the case of different tower types, the relative errors of 60 transmission towers calculated by different tilting methods and tilting degrees were compared and analyzed. On the whole, the relative errors of different types of tower increased with the increase of tilt degree. However, the relative error corresponding to the same tilting method did not fluctuate greatly with the different types of tower, and the relative error did not show obvious variation, indicating that different types of tower had little influence on the calculation accuracy of the tilt degree.

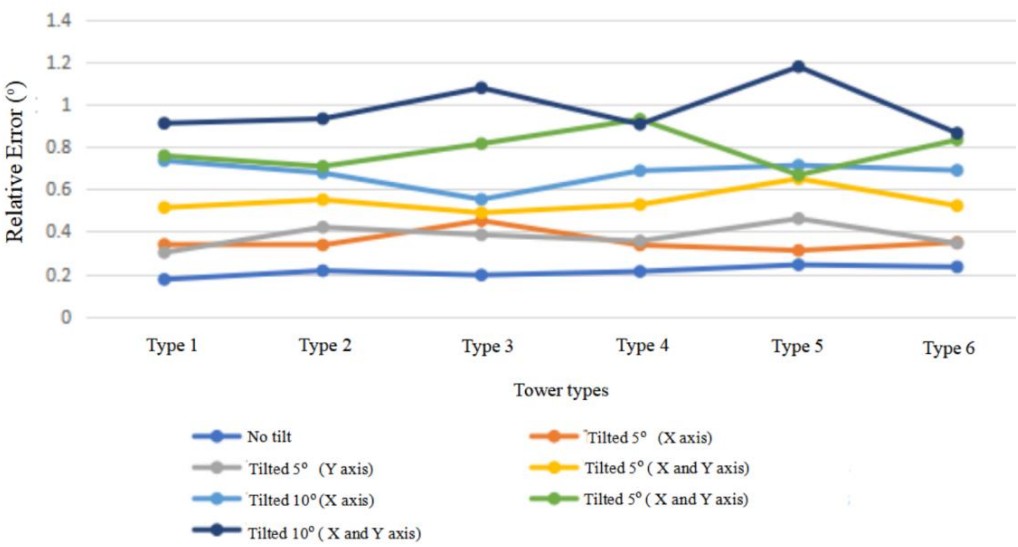

**Figure 13.** Relative error diagram of different tower types.

By analyzing the experimental results, it was found that when the tilting angle was too large, the obtained tilt degree would have too large a deviation. This was due to the inability to extract the correct feature plane because of the tilting of tower steel beams. In fact, if there is a 10° tilt of the tower, it can be found directly by visual observation, and the tower would fall with a tilt of over 10°, so the method of this paper focused on the measurement of smaller angle tilted towers.

In order to analyze the accuracy of the proposed method for measuring the tilted state of towers with a small tilt angle, the tilt simulation interval was decreased for further simulation. Specifically, the tilt angle was limited to 3°, and 1° was an interval. A total of 60 transmission towers were tilted 1°, 2°, and 3° as indicated above. The average relative error was calculated, as shown in Table 5.

**Table 5.** Tower tilt calculation [1].

| Tilt | 0° | 1° | 2° | 3° | Mean |
|---|---|---|---|---|---|
| *X* axis | 0.212 | 0.255 | 0.263 | 0.368 | 0.275 |
| *Y* axis | 0.212 | 0.287 | 0.301 | 0.392 | 0.298 |
| *X* first and then *Y* | 0.212 | 0.368 | 0.347 | 0.441 | 0.342 |
| Mean | 0.212 | 0.303 | 0.304 | 0.400 | 0.304 |

[1] These values are in degrees.

It can be seen from Table 5 that when the tilt degree was small, the errors of tower tilt measurement were all within 0.5°, which represents an accurate measurement of the tower tilt state and subtle angle changes, so as to prevent tower tilt problems.

The relationship between relative error, tilt angle, and direction is shown in Figure 14. With the increase of tilting angle, the relative error still had a slight upward trend, and the error of tilting toward the axis was relatively small compared with other directions. The point cloud elevation histogram based on the algorithm in this paper was projected along the *X*-axis direction, so the influence of the *X*-axis tilt on the accuracy was relatively small.

The experimental data were also fully consistent, which proved the effectiveness and high precision of the tower tilt evaluation algorithm.

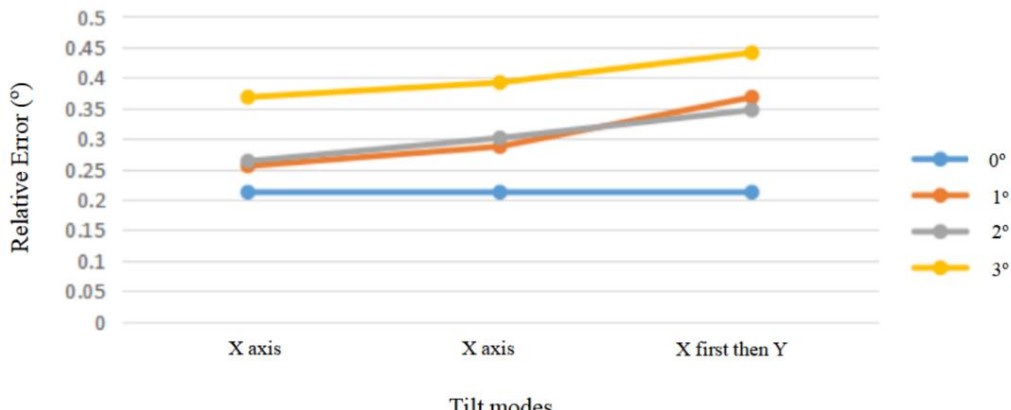

**Figure 14.** Relative error diagram of different tilting modes.

### 3.3. Analysis of Tower Head Tilt State Evaluation

In this paper, a total of 157 tower head point clouds under test and standard tower head in the model library were registered. After registration, the shape variable of tower head was calculated, and the test tower head was evaluated according to the overall deformation evaluation standard of tower head, as shown in Table 6.

**Table 6.** Evaluation results of overall deformation of tower head.

| Deformation Evaluation | Low Risk SAFE | Low Risk to Be Detected | Low Risk Need Repair | High Risk Safe | High Risk to Be Detected | High Risk Need Repair |
|---|---|---|---|---|---|---|
| Number | 152 | 4 | 0 | 1 | 0 | 0 |

It can be seen from Table 6 that among the 157 classified samples tested in this paper, 152 towers belong to the low-risk and relatively safe state, indicating that most of the towers did not undergo large deformation and the steel structure was firm. The one tower head in a high-risk and relatively safe state is caused by misclassifying the Type 2 tower head into Type 3 and using the standard Type 3 tower head in the model set for registration calculation.

In order to observe the deformation of tower head more directly, the deformation value of tower head is converted into RGB color difference for visual analysis, and the point cloud of tower head is divided into red, blue and black colors by setting the threshold value. Red represents the high-risk area with large offset, blue represents the low-risk area with moderate offset, and black represents the safe area with small offset, as shown in Figure 15.

As can be seen from Figure 15, the blue and red dots are mostly distributed in the areas connected with transmission lines or towers. Some of the red areas are in the middle of the tower, where the point cloud is not complete due to the blockage of the external tower. Therefore, the calculation results have more deviations, which is consistent with the actual situation, and also prove the effectiveness of the algorithm in this paper.

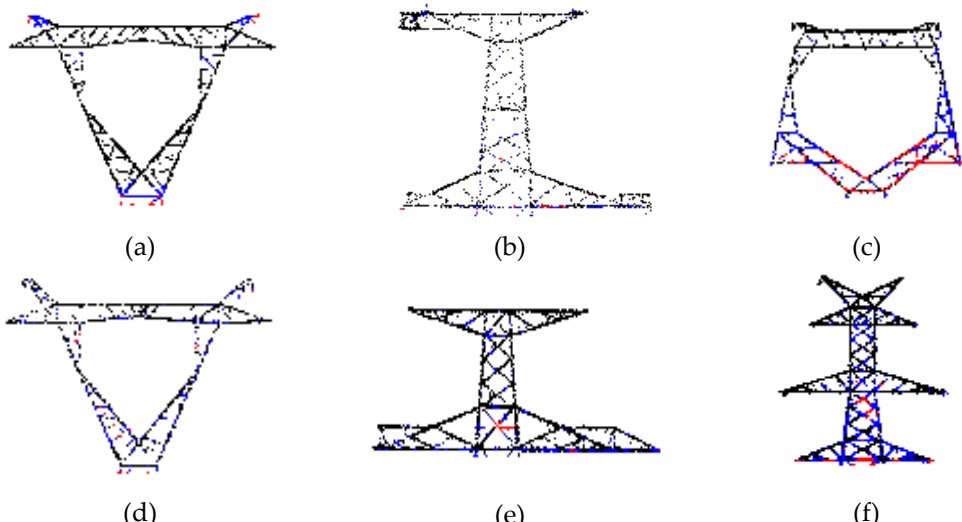

**Figure 15.** Visualization of tilt deformation using chromatic aberration for different types of tower body (**a**–**f**) represents six typical types of tower body.

## 4. Discussion

In this paper, we proposed a more targeted method to detect the tilt of the tower body and the deformation of the tower head in view of the characteristics of tilt deformation in different parts of the tower, which made the detection more clear and accurate. Compared with the traditional manual inspection, the use of UAV 3D laser scanning technology had the advantages of high efficiency, high flexibility, little environmental impact, and low risk. However, there were still some problems and improvements worth discussing.

One issue was the optimization of the filtering and extraction of the transmission tower. In the process of tower body fitting and tower head matching, it was found that the miscellaneous points around the transmission tower, such as power line point cloud and vegetation point cloud at the foot of tower, would affect the subsequent calculation process of tower, resulting in tower body extraction errors and tower head matching problems. In this paper, although most of the noise points were filtered out using the progressive iterative filtering strategy, there were still difficulties in eliminating the noise points close to the tower body. Therefore, to further improve the calculation accuracy of tilt deformation calculation, more effective point cloud filtering and extraction strategy are needed to improve the matching accuracy of tower body fitting and tower head matching.

In addition, the following two issues regarding the universality of the method need to be addressed if the research is to be applied in practice. On the one hand, although the tower body and head segmentation method based on point cloud elevation histograms worked well for the six types of tower heads studied in this paper, there were still some problems for some special towers, such as cases in which the tower head accounted for only a tenth or less of the overall length of the tower. Therefore, on the basis of the research method in this paper, further research on special towers should be carried out to achieve a new breakthrough in the generalization of all kind of towers.

On the other hand, the universality of the tower head tilt calculation was also a question worth exploring. In this paper, by matching the tower head under test with the tower head model library, we designed a tower head deformation calculation method based on the point cloud offset distance. However, this method was only applicable to the tower heads whose type could be accurately judged and which belonged to an existing tower head type in the tower head model library. This method was inflexible, and was also related to the precision of the training model. In practical application, the tilt detection was often not limited to several fixed types of tower, and some towers may have several long steel beams to reinforce. Therefore, the length, absolute angle, relative angle and

other parameters of tower head diaphragm structure can be further investigated to design appropriate calculation and evaluation methods to improve universality in the future.

## 5. Conclusions

In this paper, we proposed a method for transmission tower tilt status assessment by segmenting the tower into different parts and adopting different strategies for different parts of the tower. Based on the multi-dimensional geometric structure features of the tower 3D laser point cloud data, the tower body point cloud data were segmented from the whole tower point cloud, and the tower feature planes and elevations were extracted and combined with the elevation histogram. Next, the outer contours of the feature were extracted through the Alpha Shapes and the Pipeline algorithm. Finally, the tower body was fitted with four-prism platform, and the positioning and orientation of the tower were realized. Experimental results showed that the proposed method could accurately fit the central axis of the tilted tower within a certain angle, and the accuracy of the tilted tower angle measurement was better than 0.5 degrees. For the structurally complex tower head, the offset distance between the tower head model and the point cloud to be measured was calculated by model-driven method, and the tower head risk was evaluated by the deformation parameters.

Using UAV 3D laser scanning technology for transmission tower tilt measurement can overcome the shortcomings of traditional detection methods. This new method has the advantages of flexible take-off and landing, low altitude flight, high speed, high accuracy, and no special geographical restrictions. The automatic measurement and risk analysis scheme of transmission tower inclination based on UAV LiDAR proposed in this paper can provide the power grid operation and maintenance department with the on-site situation of transmission lines in real time, so that it can take corresponding treatment measures in time in the face of accidents, reduce the losses caused by tower inclination deformation, and significantly improve the rapid and batch intelligent perception level of transmission tower risk state, which has wide prospects for application.

**Author Contributions:** Conceptualization, Z.L. and Q.H.; methodology, S.W. and H.G.; software, Q.J.; validation, H.G., Z.L. and Q.J; investigation, Z.L.; resources, H.G.; writing—original draft preparation, Z.L.; writing—review and editing, S.W.; supervision, Q.H.; project administration, Z.L. and H.G.; funding acquisition, Z.L. and H.G. All authors have read and agreed to the published version of the manuscript.

**Funding:** This research was funded by Science and Technology Project of SGCC, grant number5500-202015074A-0-0-00.

**Institutional Review Board Statement:** Not applicable.

**Informed Consent Statement:** Not applicable.

**Data Availability Statement:** Not applicable.

**Conflicts of Interest:** The authors declare no conflict of interest.

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
