# Peer review of "A Transmission Tower Tilt State Assessment Approach Based on Dense Point Cloud from UAV-Based LiDAR"

_remotesensing, doi:10.3390/rs14020408_

Round 1
Reviewer 1 Report
I think this paper provides a novel remote sensing application with potential real world use. While I provided some proposed changes in attached version of the paper, I think it would certainly benefit from a good grammatical review.
I think the method and conclusion sections needs to be strengthened at a minimum, and this in part maybe accomplished by improving the grammar. I am not sure one could reproduce the results from the descriptions.
In addition, I think the latter sections need to address how this would be tested in real world environments, e.g. damaged towers, and what the expected false alarm/ detection rate might be, and what future steps need to be taken it the future to bring this work to a logical conclustion

Author Response
The authors would like to express their gratitude to the reviewer for his/her constructive and helpful comments for substantial improvement of this paper. The manuscript has been carefully revised according to your comments and suggestions, which are very valuable to the improvement of this manuscript. We hope our revision has improved the paper to a level of your satisfaction. The replies to your specific comments/suggestions are as follows.
Comment 1: While I provided some proposed changes in attached version of the paper, I think it would certainly benefit from a good grammatical review.
Reply: We would like to express our thanks to the reviewer for the meticulous suggestions which have a very important impact on the accuracy and rigor of our articles. We have modified our manuscripts in accordance with the reviewer’s suggestions. Since these two sections have many details to modify, we have not included the rewritten one in this response.
Comment 2: The method and conclusion sections needs to be strengthened at a minimum, and this in part maybe accomplished by improving the grammar.
Reply: Thanks for the reviewer's comments. We are sorry for our lack of rigorous expression. We have reviewed and edited our manuscripts again. Since these two sections have many details to modify, we have not included the rewritten one in this response. We hope our revision has improved the paper to a level of your satisfaction.
Comment 3: The latter sections need to address how this would be tested in real world environments, e.g. damaged towers, and what the expected false alarm/ detection rate might be, and what future steps need to be taken it the future to bring this work to a logical conclusion.
Reply: Thanks for the reviewer's comments.
Revision in our paper: (the revisions are marked in red)

Reviewer 2 Report
My comments and questions are listed below:
- Please add a figure to illustrate the parameters used in section 2.2.3. and 2.2.4, which can help readers better understand these key factors.
- The tilt simulation in section 3.1 is unconvincing to prove the proposed algorithm could be used in real scenes. First of all, the tilting should be rotated from the base not the centroid of the tower body. Secondly, the shear, bending, and torsional deformation were not considered, which are crucial due to wind load on arm, tower, and cable. Thirdly, the quantitative evaluation of the deformation of the tower head is missing. Please take the case of high risk but safe for example, and explain why 0.5 is the suggested threshold.
- The scanned points cloud of the transmission power tower normally includes the deflections due to the weight of the cable and tower itself. The deformation can be estimated using structural analysis tools (e.g. sap2000). Therefore, the designed tilt angle should be eliminated from the mapping process. Moreover, the critical range of the tilting angle along the x and y major axis can be decided with supportive information.
- Most of the breaking points are yielded in certain sections, e.g. the connecting layer of the tower head and body. How do you handle this kind of deformation? The body and base are straight but heavily tilted on the connecting layer.
Author Response
Replies to Reviewer 2
The authors would like to express their gratitude to the reviewer 2 for his/her constructive and helpful comments for substantial improvement of this paper. The manuscript has been carefully revised according to your comments and suggestions, which are very valuable to the improvement of this manuscript. We hope our revision has improved the paper to a level of your satisfaction. The replies to your specific comments/suggestions are as follows.
Comment 1: Please add a figure to illustrate the parameters used in section 2.2.3. and 2.2.4, which can help readers better understand these key factors.
Reply: Thanks for the reviewer's comments. We have supplemented and redrawn several figures to illustrate the parameters. Hope that these figures can better show the structure and working principle of our method.
Revision in our paper: (the revisions are marked in red)
L96:
Figure 9. Central axis
(a) (b)
Figure 10. Central axis projection
Figure 11. Schematic diagram of tower tilt angle calculation
Comment 2: The tilt simulation in section 3.1 is unconvincing to prove the proposed algorithm could be used in real scenes. First of all, the tilting should be rotated from the base not the centroid of the tower body. Secondly, the shear, bending, and torsional deformation were not considered, which are crucial due to wind load on arm, tower, and cable. Thirdly, the quantitative evaluation of the deformation of the tower head is missing. Please take the case of high risk but safe for example, and explain why 0.5 is the suggested threshold.
Reply: Thanks for the reviewer's comments. We are sorry for the inadequate and unclear statement in the previous manuscript. Firstly, the simulated data were generated by rotating certain angles around different axes with the base center as the origin. We have supplemented the explanation in the manuscript. For the second issue, we admit that the proposed method can only calculate the tilt angle of the tower body and cannot calculate the torsional deformation of the tower body. In practice, however, the shear, bending, and torsional deformation occurred mainly in the tower head section. For the third question, in fact, we have used the unprocessed tower head in the raw data as the experimental data set for the tower head tilt and deformation analysis, since the original tower head had slight deformation due to uneven force caused by power line pulling and influence of wind. We hope that these changes will make our manuscript more complete and logical.
Revision in our paper: (the revisions are marked in red)
L246: " In the data set, most of the transmission towers were vertical, and the real tilted towers were rare. To verify the accuracy of the method for calculating the tower body tilt, simulations were carried out on the collected sample data to obtain towers with different tilt angles. Specifically, 10 towers were randomly selected from each category, and the simulated data were divided into six groups, each of which took the base center as the origin and rotated a certain angle around a different axis, as shown in Table 3. ”
Comment 3: The scanned points cloud of the transmission power tower normally includes the deflections due to the weight of the cable and tower itself. The deformation can be estimated using structural analysis tools (e.g. sap2000). Therefore, the designed tilt angle should be eliminated from the mapping process. Moreover, the critical range of the tilting angle along the x and y major axis can be decided with supportive information.
Reply: The tower structure design usually has strict mechanical requirements. The tower structure is assumed as a rigid whole, especially the tower body structure is stable and has sufficient overall strength. The power tower inclination state evaluation with LiDAR data proposed in this paper is to divide the tower into two parts. The inclination angle of the tower body is measured by fitting the tower center-line, without considering the local nonlinear deformation caused by different forces. While, the structure of the tower head is relatively unstable compared with the tower body and is greatly affected by the environment such as power line and wind, In this paper, the accurate shape comparison analysis with the designed standard tower head model is used to evaluate its state. The structural mechanics analysis in your suggestion to analyze and evaluate from the perspective of structural mechanics, while the method in this paper is to directly measure the deformation at different parts. The later research can analyze the correlation between the two, so as to provide an accurate reference for the design and maintenance of tower structure.
Comment 4: Most of the breaking points are yielded in certain sections, e.g. the connecting layer of the tower head and body. How do you handle this kind of deformation? The body and base are straight but heavily tilted on the connecting layer.
Reply: Thanks for the reviewer's comments. We refer to the connecting layer as the tower shoulder in the paper. As can be seen in Figure 1a and Figure 13, after the segmentation of the tower body and tower head, the connecting layer belongs to the tower head. Therefore, on the one hand, the deformation of connecting layer does not affect the tower body tilt calculation; on the other hand, the deformation status of connecting layer can be determined at section 2.4 Tower head tilt status evaluation.

Reviewer 3 Report
The authors present an approach for assessing the inclination of transmission towers, which is based on point clouds from UAV-based LiDAR. The paper is interesting and I suggest publishing it after a few reviews to address the following comments:
- Some parts of the manuscript appear heavily fragmented (i.e. Sections 2, 3 and 4) because the authors use many subparagraphs and bullet points.
- Details on the point clouds should be needed (e.g. method of obtaining point clouds, co-registration errors, etc.)
- Section 4 "Discussion" appears to be a summary of the results, a more critical review should be provided.
Author Response
Replies to Reviewer 3
The authors would like to express their gratitude to the reviewer 3 for his/her constructive and helpful comments for substantial improvement of this paper. The manuscript has been carefully revised according to your comments and suggestions, which are very valuable to the improvement of this manuscript. We hope our revision has improved the paper to a level of your satisfaction. The replies to your specific comments/suggestions are as follows.
Comment 1: Some parts of the manuscript appear heavily fragmented (i.e. Sections 2, 3 and 4) because the authors use many subparagraphs and bullet points.
Reply: Thanks for the reviewer's detailed comments. We have significantly modified the structure. We hope that these changes will improved the paper to a level of your satisfaction.
Comment 2: Details on the point clouds should be needed (e.g. method of obtaining point clouds, co-registration errors, etc.)
Reply: Thanks for the reviewer's comments. We are very sorry for the incomplete statements here. We have supplemented some related details in the manuscript.
Revision in our paper: (the revisions are marked in red)
L396: “In the paper, the feasibility of the proposed method was verified using the 3d laser point cloud data provided by Shanxi Power Grid Company. The data were collected using the UAV equipped with a VUX-1 LiDAR sensor. The experimental data set contained 157 transmission towers of 6 categories, as shown in Table 2. The point cloud density was about 100 pts/m2. The number of point clouds varied considerably across the 6 types of transmission towers, with the minimum point being 4,000 and the maximum point being 40,000 and the average point being 20,000.”
Comment 3: Section 4 "Discussion" appears to be a summary of the results, a more critical review should be provided.
Reply: Thanks for the reviewer's comments. We have pointed out the shortcomings of the paper and gave some suggestions for future research, as shown in the manuscript and below. Hope that the revised “Discussion” section can meet with approval.
Revision in our paper: (the revisions are marked in red)
L493: In this paper, Method in this study for the tower tilt deformation detection provides a new train of thought. Compared with general tower tilt detection methods, we proposed a more targeted method to detect the tilt of the tower body and the deformation of the tower head in view of the characteristics of tilt deformation in different parts of the tower, which made the detection more clear and accurate. Compared with the traditional manual inspection, the use of UAV 3D laser scanning technology had the advantages of high efficiency, high flexibility, little environmental impact and low risk. We have studied some links in the measurement technology of transmission tower tilt state, However, there are were still some problems and improvements worth discussing.
4.1. Point Cloud Filtering and Extraction Optimization of Transmission Towers
The first issue was the optimization of the filtering and extraction of transmission tower. In the process of tower body fitting and tower head matching, it was found that the miscellaneous points around the transmission tower, such as power line point cloud and vegetation point cloud at the foot of tower, would affect the subsequent calculation process of tower, resulting in tower body extraction errors and tower head matching problems. In this paper, although most of the noise points were filtered out using the progressive iterative filtering strategy, there were still difficulties in eliminating the noise points close to the tower body. Therefore, to further improve the calculation accuracy of tilt deformation calculation, more effective point cloud filtering and extraction strategy were needed to improve the matching accuracy of tower body fitting and tower head matching.
In addition, the following two issues regarding the universality of the method need to be addressed if the research is to be applied in practice. On the one hand, although the tower body and head segmentation method based on point cloud elevation histograms had worked well for the six types of tower heads studied in this paper, but there were still some problems for some special towers, such as the tower head accounted for only a tenth or less of the overall length of the tower. Therefore, on the basis of the research method in this paper, further research on special towers should be carried out to achieve a new breakthrough in the generalization of all kind of towers.
4.2. More General Tilt Deformation Method for Tower Head
On the other hand, the universality of the tower head tilt calculation was also a question worth exploring. In this paper, by matching the tower head under test with the tower head model library, we designed a tower head deformation calculation method based on the point cloud offset distance. However, this method was only applicable to the tower head under test which can be accurately judged the type and belonged to the existing tower head type in the tower head model library. This method was inflexible, and was also related to the precision of the training model. In practical application, the tilt detection was often not limited to several fixed types of tower models, and some towers may have several long diaphragms steel beams to reinforce. Therefore, the length, absolute angle, relative angle and other parameters of tower head diaphragm structure can be further investigated to design appropriate calculation rules and evaluation methods to improve universality in the future.

Round 2
Reviewer 2 Report
Follow up on my previous review.
C1. Figure 9 to 11 clearly illustrated the parameters used in section 2.3.
C2-4. In section 2.2.3, the proposed algorithm claimed the ability to find the contours of each cross section. With the corner points and the centroid should be able to estimate the torsional deformation. I wish to see more complete research and experimental results with real deformation data. The simulation of the tilt experiment (section 3.1) simplified the rotation of the tower body as a rigid body, which is not practical in real scenes. The threshold of the tower head overall risk status (0.5) is defined without convincing reasons. Please give some examples to verify your hypothesis.
Author Response
The authors would like to express their gratitude to the reviewer for his/her constructive and helpful comments for substantial improvement of this paper. The manuscript has been carefully revised according to your comments and suggestions, which are very valuable to the improvement of this manuscript. We hope our revision has improved the paper to a level of your satisfaction. The replies to your specific comments/suggestions are as follows.
Comment 1: C1. Figure 9 to 11 clearly illustrated the parameters used in section 2.3.
Reply: Thanks for the reviewer's comments.
Comment 2: C2-4. In section 2.2.3, the proposed algorithm claimed the ability to find the contours of each cross section. With the corner points and the centroid should be able to estimate the torsional deformation. I wish to see more complete research and experimental results with real deformation data. The simulation of the tilt experiment (section 3.1) simplified the rotation of the tower body as a rigid body, which is not practical in real scenes. The threshold of the tower head overall risk status (0.5) is defined without convincing reasons. Please give some examples to verify your hypothesis.
Reply: Thanks for your comments. We have added some practical data we just obtained from real project for the tilt angle calculation (Section 3.4).
The threshold of the tower head overall risk status (0.5) is from the engineering experience. This is a preliminary attempt to establish the relationship between geometric deformation and actual risk, which needs more engineering data and expert experience and knowledge in the follow-up. In fact, the geometric deformation between the actual tower head and the standard model calculated by the method in this paper is an overall shape variable, and its distribution in different parts is not considered, which is also the content that needs further research of risk assessment in the future. We have added some words about the risk status threshold
Revision in our paper: (Section 3.4 and Line 396-401 )

This manuscript is a resubmission of an earlier submission. The following is a list of the peer review reports and author responses from that submission.